# The Peptide AWRK6 Alleviates Lipid Accumulation in Hepatocytes by Inhibiting miR-5100 Targeting G6PC

**DOI:** 10.3390/ijms242216141

**Published:** 2023-11-09

**Authors:** Jiaxin Liu, Ying Liu, Qiuyu Wang, Lili Jin, Dianbao Zhang

**Affiliations:** 1School of Life Sciences, Liaoning University, Shenyang 110036, China; lisaliu11178@163.com (J.L.); qiuyuwang@lnu.edu.cn (Q.W.); 2Department of Stem Cells and Regenerative Medicine, Key Laboratory of Cell Biology, National Health Commission of China, and Key Laboratory of Medical Cell Biology, Ministry of Education of China, China Medical University, Shenyang 110122, China; yingliu@cmu.edu.cn

**Keywords:** NAFLD, MAFLD, lipid accumulation, peptide, AWRK6, miR-5100, G6PC

## Abstract

Non-alcoholic fatty liver disease (NAFLD) is the leading chronic liver disease, with a worldwide prevalence of more than 25%, and there is no approved drug for NAFLD specifically. In our previous study, the synthetic peptide AWRK6 was found to ameliorate NAFLD in mice. However, the mechanisms involved are still largely unknown. Here, AWRK6 treatment presented an alleviative effect on lipid accumulation induced by oleic acid in hepatocytes. Meanwhile, miR-5100 and miR-505 were found to be elevated by oleic acid induction and reversed by AWRK6 incubation. Further, the miR-5100 inhibitor inhibited oleic acid-induced lipid accumulation, and the alleviation effect of AWRK6 was partially counteracted by miR-5100 mimics. The screening of potential target genes revealed that a catalytic subunit of G6Pase G6PC was significantly inhibited by miR-5100 mimics transfection in both mRNA and protein levels. The direct targeting of miR-5100 on G6PC was verified by a Dual-Luciferase Reporter Assay. Moreover, the mRNA and protein levels of G6PC were found to be significantly increased by AWRK6 treatment. These results suggested that the peptide AWRK6 could alleviate lipid accumulation in hepatocytes, partly through reducing miR-5100 to restore one of its targets: G6PC. Thus, AWRK6 has the potential to treat NAFLD. Additionally, miR-5100 is a mediator of lipid accumulation in hepatocytes, which could be targeted by AWRK6.

## 1. Introduction

Non-alcoholic fatty liver disease (NAFLD) is the most prevalent liver disease, affecting more than one quarter of the global population, and its prevalence is continuously increasing [1]. In the past 3 years, metabolic-associated fatty liver disease (MAFLD) has been suggested as a replacement name for NAFLD, and this has been endorsed by various associations and experts [2]. Lifestyle interventions including healthy diet and regular exercise are the primary recommendations for the prevention and treatment of NAFLD, but the long-term sustainability of these interventions is poor due to multiple social, physiological, and psychological reasons. To prevent NAFLD’s progression to non-alcoholic steatohepatitis (NASH), cirrhosis, hepatocellular carcinoma (HCC), and death, various therapeutic agents targeting multiple potential pathways are under investigation. As of now, there are still no approved drugs for the prevention and treatment of NAFLD [2,3,4]. Therefore, there is an urgent need to develop novel drug candidates and to investigate their pharmacological mechanisms.

miRNA is a type of small non-protein-coding single-strand RNA around 20–25 nucleotides in length. Generally, miRNA binds to the 3′-untranslation region (3′-UTR) of target mRNA to repress gene translation or trigger mRNA degradation. Furthermore, some miRNAs work as gene activators, and some were found to bind to coding region 5′-UTR and its receptors. One miRNA regulates numerous targets and one mRNA is targeted by various miRNAs [5]. In the liver, with NAFLD, changes in miRNA expression patterns are associated with and involved in the progression of NAFLD. MiRNAs act as messengers for intercellular communication in hepatocytes, stellate cells, and immune cells, among others [5,6]. Thus, miRNAs are promising potential biomarkers and important therapeutic targets for the diagnosis and treatment of NAFLD, respectively.

AWRK6 is a linear peptide developed based on Dybowskin-2CDYa, a natural peptide identified in the skin of the Chinese frog *Rana dybowskii* in our previous study. In high-energy diet-induced NAFLD mice, obesity and hepatic steatosis were significantly alleviated by the intraperitoneal administration of AWRK6, through relieving the abnormal lipid metabolism [7]. In addition, RNA-seq data revealed that some miRNAs, including miR-5100 and miR-505, were involved in the regulation of metabolism in our previous work [8]. However, the mechanisms involved are still largely unknown. In the present study, the alleviative effects of AWRK6 on lipid accumulation in hepatocytes were evaluated. Further, miR-5100 was found to be inhibited by AWRK6, and G6PC was found to be a downstream target of miR-5100.

## 2. Results

### 2.1. AWRK6 Alleviated Lipid Accumulation Induced by Oleic Acid

To evaluate the effects of AWRK6 on the lipid accumulation in hepatocytes, an oleic acid-induced HepG2 cell steatosis model was utilized. The HepG2 cells were incubated with oleic acid at 0.3, 0.4, 0.5, 1.0, and 1.5 mM for 24 h. The triglyceride in HepG2 cells was determined using a Triglyceride Colorimetric Kit, and the cell viability was analyzed using a Cell Counting Kit-8 (CCK-8) assay. Upon oleic acid treatment, the triglyceride was increased and the cell viability was decreased, both in a concentration-dependent manner (Figure 1A,B). The triglyceride was increased by 100% and the cell viability was decreased by 20% at a concentration of 0.5 mM, which was selected for the consequent experiments. As shown in Figure 1C,D, the HepG2 cells were incubated with oleic acid for 24 h, and then treated with the synthetic AWRK6 (0.01, 0.05, 0.10, 0.50, and 1.00 μM) for another 24 h. The triglyceride in HepG2 cells was significantly inhibited by AWRK6 at all the concentrations in a concentration-dependent manner. Meanwhile, the cell viabilities were not changed upon AWRK6 treatment. These results indicated the alleviation effects of AWRK6 on oleic acid-induced lipid accumulation in HepG2 hepatocytes.

### 2.2. AWRK6 Reduced Oleic Acid-Elevated miR-5100 and miR-505

The HepG2 cells were incubated with 0.5 mM oleic acid for 24 h and then treated with 0.5 μM AWRK6 for another 24 h. The miRNA expression changes in HepG2 cells upon oleic acid induction and AWRK6 treatment were analyzed using real-time PCR. As presented in Figure 2, the real-time PCR results revealed that miR-5100 and miR-505 were upregulated by oleic acid induction, and the following AWRK6 treatment significantly reduced their expression, indicating they are potential mediators of the alleviation of lipid accumulation by AWRK6.

### 2.3. miR-5100 Mediated Alleviation of Lipid Accumulation by AWRK6

The functions of miR-5100 and miR-505 in metabolic regulation are not yet clear. miRNA mimics and inhibitors were separately transfected into HepG2 cells to achieve gain of function and loss of function. The real-time PCR results in Figure 3A–D showed a significant increase and decrease in miR-5100 and miR-505 levels at 24 h after transfection, respectively. Additionally, in oleic acid-induced HepG2 cells, triglyceride was reduced by miR-5100 inhibitors’ transfection for 24 h (Figure 3E). In addition, the triglyceride level was not altered after miR-505 mimics’ or inhibitors’ transfection (Appendix A). Further, the AWRK6-reduced triglyceride in oleic acid-induced HepG2 cells was found to be reversed partly by miR-5100 mimics’ transfection (Figure 3F). These data suggest that miR-5100 contributed to lipid accumulation in hepatocytes, and that it mediated the alleviation of lipid accumulation by AWRK6.

### 2.4. G6PC Was a Direct Target of miR-5100

The target genes of miR-5100 were predicated using the intersection of miRDB and TargetScan databases, with reference to NAFLD-related genes in the Disease database. As presented in Figure 4A–C, G6PC was not changed by oleic acid induction, and AWRK6 treatment elevated G6PC at mRNA and protein levels. Then, the mRNA level of G6PC was analyzed after the transfection of miR-5100 mimics into the HepG2 cells using real-time PCR, which revealed that the mRNA level of G6PC was significantly reduced by miR-5100 mimics transfection (Figure 4D). The protein levels of G6PC after the transfection of miR-5100 mimics were further verified using Western blotting (Figure 4E,F). As shown in Figure 4G, there are two predicated conserved targeting sites of miR-5100 in the 3′-UTR of G6PC. The recombinant luciferase reporter plasmids were constructed by inserting the wildtype and mutant targeting sequences into the downstream of the reporter gene luciferase. Using a Dual-Luciferase Reporter Assay, the luciferase activity in the cells transfected with miR-5100 mimics and the recombinant plasmid containing wildtype targeting site 2 was found to be significantly reduced, indicating that miR-5100 targeted the 3′-UTR of G6PC mRNA. Further, the mRNA and protein levels of G6PC in HepG2 cells upon oleic acid induction and AWRK6 treatment were analyzed. These data suggested G6PC was a direct target of miR-5100, which mediated the alleviation of lipid accumulation in hepatocytes by AWRK6.

## 3. Discussion

NAFLD is characterized by lipid accumulation in the liver without secondary causes, including significant alcohol consumption and other attributable conditions [9]. Absorption and de novo lipogenesis are the main sources of liver fatty acid, which is eliminated by β-oxidation or forms triglyceride through glycerol esterification. The triglyceride can be released into circulation as very low-density lipoprotein (VLDL) or temporarily stored in hepatocytes. The excessive generation or impaired release of triglyceride leads to the formation of lipotoxic lipid, promoting the progression of NAFLD.

Subsequently, various treatment strategies aim to reduce lipid accumulation in hepatocytes [3]. Recently, we found that the peptide AWRK6 alleviated NAFLD in mice, which was induced by a high-energy diet. Here, the in vitro experiments further suggested that AWRK6 alleviated lipid accumulation in hepatocytes. Considering the important regulatory roles of miRNAs in lipid accumulation, the effects of AWRK6 treatment on miRNA expression in oleic acid-induced hepatocytes were analyzed. miR-5100 and miR-505 were found to be upregulated in oleic acid-induced hepatocytes, and their levels were significantly downregulated by AWRK6 treatment. Moreover, miR-5100 overexpression partially counteracted the mitigating effect of AWRK6 on lipid accumulation. Further, G6PC was found to be a direct target gene of miR-5100. These data suggest that AWRK6 alleviates oleic acid-induced lipid accumulation in hepatocytes by inhibiting miR-5100, in which G6PC is a target gene involved with miR-5100.

G6PC is a catalytic subunit of G6Pase, which catalyzes the hydrolysis of glucose-6 phosphate (G6P) to produce glucose. The deficiency in G6PC causes G6P accumulation in the liver and kidneys, leading to the dysregulation of metabolism [10]. In obese patients, G6PC expression was found to be reduced by half in patients with NAFLD versus those without NAFLD [11]. Another investigation revealed that the G6PC expression was lower in NASH patients who consumed a high amount of carbohydrates [12]. The liver-specific G6PC knockout mice developed glycogen storage disease type Ia (GSDIa), exhibiting NAFLD features [10]. In this study, G6PC expression could be elevated by AWRK6 treatment in hepatocytes with lipid accumulation via reducing miR-5100. It is important to note that the development of metabolic diseases is complex. The downregulation of G6PC in patients with NAFLD limits the increase in blood glucose. Additionally, the restoration of G6PC is likely to lead to an increase in blood glucose, which, in turn, increases the risk of diabetes, etc. The dysregulation of G6P is a hallmark of abnormal metabolic diseases including NAFLD, and combining restoration of G6PC with other interventions might provide novel therapeutic strategies for the treatment of NAFLD.

This study explores the mechanism of AWRK6 at the cellular and molecular levels based on the previous finding that AWRK6 alleviates NAFLD in mice through regulating lipid metabolism. Considering the important roles of miRNAs in the development of NAFLD, we started to explore the possible mechanisms from the miRNA perspective. Here, we were fortunate to discover that miR-5100 might mediate the mitigating effects of AWRK6 on lipid accumulation. However, these data only provide a preliminary miRNA perspective, and much remains unclear. During this study, we attempted to detect G6PC mRNA and protein expression levels in HepG2 transfected with miR-5100 mimics, and additionally supplemented with OA and AWRK6. The changes were not statistically significant. Therefore, we speculate that G6PC is one of the mediators of AWRK6 in alleviating lipid accumulation. The other mechanisms involved are to be further investigated in depth in subsequent studies. It is also possible that we have not found suitable experimental conditions to show the changes in G6PC. In addition, miRNAs have a wide range of roles, can be delivered between different cells and tissues, and are tissue- and cell-specific for functions, making the data on miRNAs difficult to translate to the clinic. It is challenging to localize the tissue of origin of miRNAs; much work is needed to explain the disease association to microRNAs’ regulation. This is why it is important to investigate the mode of miRNAs in depth in order to elucidate the regulatory role of miRNAs in disease development and intervention.

Notably, many miRNA studies have focused on their interactions with target genes, miRNA expression patterns under pathological conditions, and delivery between cells or tissues via exosomes and other means. The many-to-many relationship between miRNAs and target genes makes it difficult for classical research strategies to fully reveal the relationship between miRNA regulation and disease. Further combinations of classical molecular biology research strategies with high-throughput technologies and big data analysis techniques may provide better explanations.

In summary, the peptide AWRK6 was found to alleviate lipid accumulation in hepatocytes, partly through inhibiting the expression of miR-5100 to restore one of its targets: G6PC. These findings indicate that AWRK6 is a potential peptide for intervention in metabolism dysregulation, including NAFLD, and that miR-5100 is a regulator of glucose production that targets G6PC.

## 4. Materials and Methods

### 4.1. Cell Culture

The HepG2 cells (Procell, Wuhan, China) and 293T (a gift from Ziwei Miao) were maintained with DMEM/high glucose (Hyclone, Shanghai, China) and supplemented with 10% fetal bovine serum (DearyTech, Shanghai, China), 1% Penicillin-streptomycin solution (Meilunbio, Dalian, China), 1% Amphotericin B Solution (Biological Industries, Haemek, Israel), and 0.1 μg/mL Mycoplasma Removal Agent (Beyotime, Shanghai, China) in an incubator at 37 °C with 5% CO_2_. Lipo6000 Transfection Reagent (Beyotime) was used for the transfection of miRNA mimics and inhibitors following the manufacturer’s instructions. The miRNA mimics and inhibitors were chemically synthesized by GenePharma (Shanghai, China).

### 4.2. Cell Viability Assay

Cell viability was analyzed using the CCK-8 reagent (Meilunbio). The HepG2 cells were seeded into 96-well plates (8000 cells per well). After incubating overnight, the cells were treated as indicated. Then, 10 μL of the CCK-8 reagent was added into each well, and the cells were incubated at 37 °C for 1 h. The OD450 values were obtained using a microplate reader (iMARK, Bio-Rad, Hercules, CA, USA), with a reference wavelength of 630 nm.

### 4.3. Triglyceride Determination

The triglyceride content was determined using a Tissue Cell Triglyceride Content Enzyme Assay Kit (Applygen, Beijing, China). Briefly, following the manufacturer’s instructions, the cells were washed with PBS and lysed using a lysis buffer. They were then centrifuged at 2000× *g* to collect the supernatant. Then, 10 μL supernatant and 190 μL working solution were added into 96-well plates and incubated at 37 °C for 15 min. The absorbance at 560 nm was analyzed using an iMARK microplate reader.

### 4.4. Real-Time PCR

The miRNA expression and the mRNA level of predicated target genes were determined using real-time PCR. The total RNA was isolated using an RNAiso Plus (Takara, Dalian, China) following the manufacturer’s instructions, and quantified using Nanodrop 2000C (Thermo, Shanghai, China). For miRNAs, the cDNA was prepared using GoScript Reverse Transcriptase (Promega, Beijing, China). For predicated target genes, a RevertAid First Strand cDNA Synthesis Kit (Thermo) was used to prepare the cDNA. The real-time PCR was carried out with an ABI7500 Real-Time PCR System (Thermo) using PowerUp SYBR Green Master Mix (Thermo). U6 and β-actin were used as internal controls for miRNA and mRNA, separately. The relative changes in miRNA and mRNA were calculated using the 2^−ΔΔCt^ method. The primer sequences are listed in Table 1.

### 4.5. Western Blotting

The cell lysate was prepared using a RIPA buffer (Beyotime) and quantified using a BCA assay kit (Takara). Western blotting was carried out as described in our previous study. Briefly, equal amounts of total proteins were subjected to sodium dodecyl sulfate-polyacrylamide gel electrophoresis (SDS-PAGE) and transferred to polyvinylidene fluoride (PVDF) membranes. The PVDF membranes were blocked with 5% skimmed milk and incubated with primary antibodies at 4 °C overnight. The membranes were washed and incubated with HRP-conjugated secondary antibodies at room temperature for 1 h. The primary antibodies against G6PC (1:3000), β-actin (1:10,000), and the HRP-conjugated secondary antibodies (1:10,000) were all purchased from Proteintech (Wuhan, China). The protein bands were visualized using a BeyoECL Plus reagent (Beyotime) and imaged with a Tanon-5200 chemiluminescence detection system (Tanon, Shanghai, China). The images were analyzed using ImageJ software 1.46.

### 4.6. Dual-Luciferase Reporter Assay

The target genes of miRNA were predicated by miRDB and TargetScan. A Dual-Luciferase Reporter Assay was applied to verify the interaction between miRNA and the target gene. The recombinant pmirGLO (Promega, Beijing, China) vectors containing wild type or mutant target sites were constructed as described previously, using the restriction endonucleases SacI and XhoI. The sequences of oligonucleotides are listed in Table 2. The 293T cells were seeded into 24-well plates and co-transfected with vector and miRNA mimics/NC. The relative luciferase activities were analyzed using a Dual Glo Luciferase Assay System (Promega, Madison, WI, USA) on a GloMax 96 Microplate Luminometer (Promega), following the manufacturer’s instructions.

### 4.7. Statistical Analysis

Data are presented as mean ± SD. A Student’s t-test was used to analyze the difference between two groups, and a one-way ANOVA was used for more than three groups. *p* < 0.05 was considered statistically significant.

## Figures and Tables

**Figure 1 ijms-24-16141-f001:**
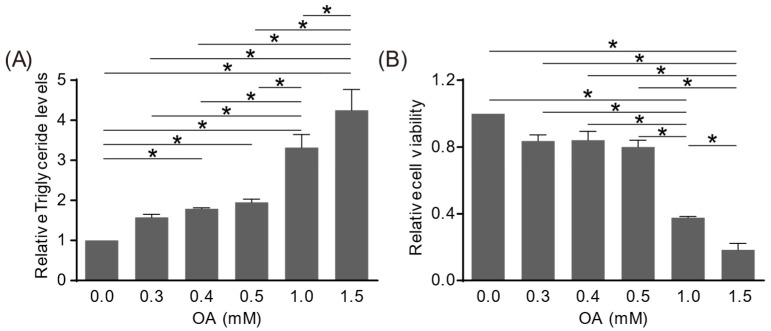
AWRK6 alleviated oleic acid-induced lipid accumulation in HepG2 cells. (**A**,**B**) The HepG2 cells were treated with oleic acid (OA) for 24 h. The triglyceride was analyzed using a Triglyceride Colorimetric Kit and the cell viability was analyzed using a CCK-8 assay. (**C**,**D**) The oleic acid-induced HepG2 cells were treated with AWRK6, then the triglyceride, and cell viability was analyzed. * *p* < 0.05, compared with indicated groups.

**Figure 2 ijms-24-16141-f002:**
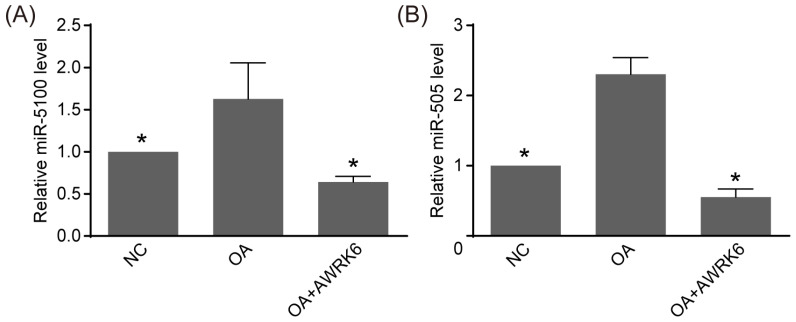
AWRK6 inhibited miR-5100 and miR-505 in oleic acid-induced HepG2 cells. (**A**,**B**) The HepG2 cells were induced using oleic acid (OA) and then treated with AWRK6. The expression of miR-5100 and miR-505 was analyzed using real-time PCR. * *p* < 0.05, compared with OA groups.

**Figure 3 ijms-24-16141-f003:**
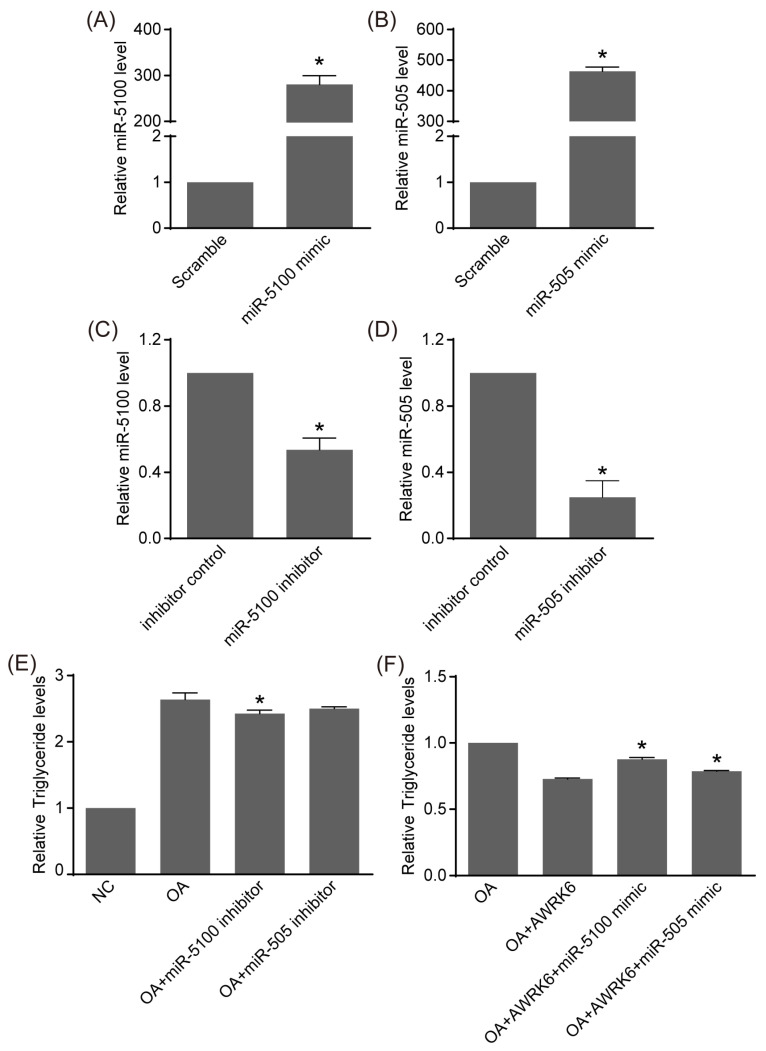
miR-5100 contributed to the alleviation of lipid accumulation via AWRK6. (**A**–**D**) The HepG2 cells were transfected with miRNA mimics or inhibitors, and the transfection efficiency was analyzed using real-time PCR. (**E**) The HepG2 cells were treated with oleic acid (OA) and transfected with miR-5100 inhibitors and miR-505 inhibitors, then triglyceride levels were analyzed. (**F**) The cells were transfected with miR-5100 mimics and miR-505 mimics, induced by oleic acid, and treated with AWRK6. The triglyceride was determined to clarify the involvement of miR-5100 in the alleviation of lipid accumulation by AWRK6. * *p* < 0.05, compared with OA groups.

**Figure 4 ijms-24-16141-f004:**
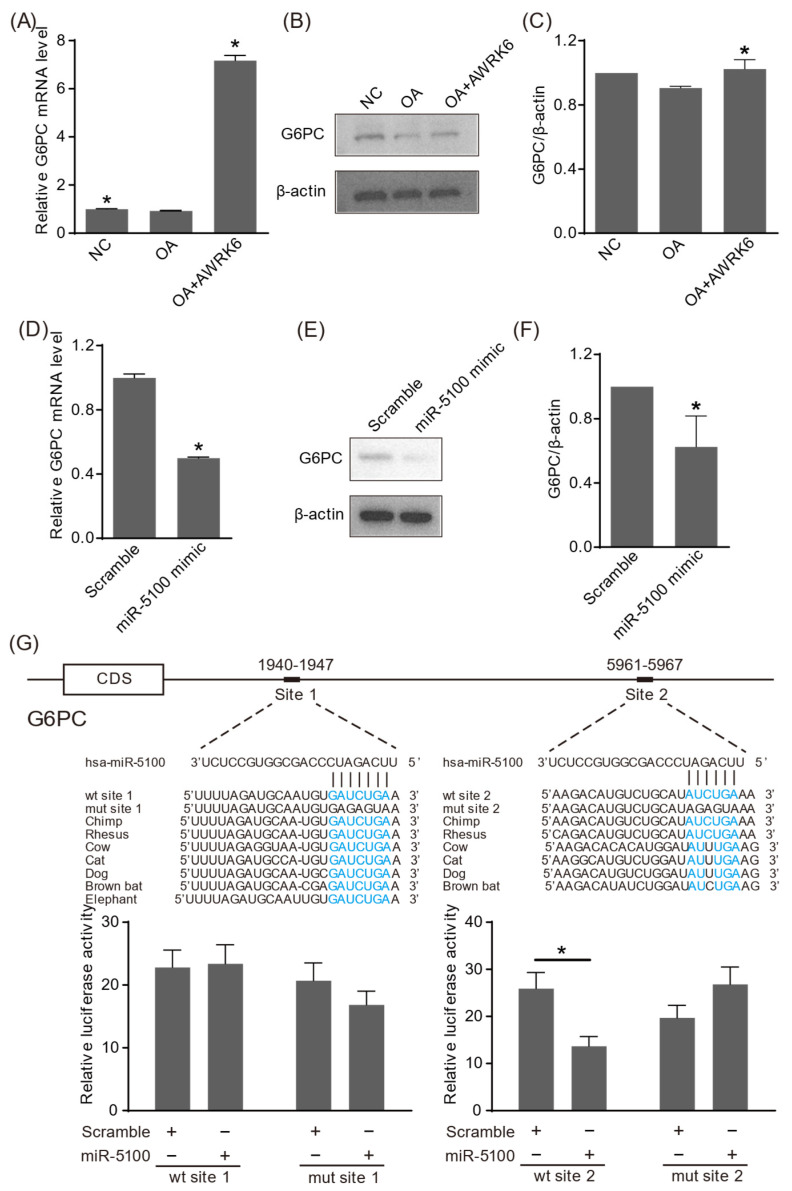
G6PC was a direct target of miR-5100. (**A**,**B**) The HepG2 cells were induced by oleic acid (OA) and treated with AWRK6, and the mRNA and protein levels of G6PC in HepG2 cells were determined via real-time PCR and Western blotting. * *p* < 0.05, compared with OA groups. (**C**) The images obtained via Western blotting in Figure 4F were analyzed using ImageJ software 1.46. * *p* < 0.05, compared with OA group. (**D**) The mRNA level of G6PC in HepG2 cells upon miR-5100 mimics’ transfection was analyzed using real-time PCR. * *p* < 0.05, compared with the scramble group. (**E**) The protein level of G6PC was detected using Western blotting. (**F**) The images obtained through Western blotting in Figure 4B were analyzed using ImageJ software 1.46. * *p* < 0.05, compared with the scramble group. (**G**) The direct targeting of G6PC by miR-5100 was analyzed using a Dual-Luciferase Reporter Assay. There were two predicated targeting sites in the 3′-UTR of G6PC. The conserved sequences are shown in blue. The cells were co-transfected with the reporter plasmids containing wildtype/mutant target sequences and miR-5100 mimics/scramble, and the relative luciferase activities were analyzed 48 h after transfection. * *p* < 0.05, compared with scramble groups.

**Table 1 ijms-24-16141-t001:** The primer sequences for real-time PCR.

Name	Sequence (5′-3′)
miR-5100 RT	GTCGTATCCAGTGCAGGGTCCGAGGTATTCGCACTGGATACGACGGTACA
miR-5100 forward	GCTTCAGATCCCAGCGGT
miR-505 RT	GTCGTATCCAGTGCAGGGTCCGAGGTATTCGCACTGGATACGACAGGAAA
miR-505 forward	GCCGTCAACACTTGCTGG
miRNA reverse	AGTGCAGGGTCCGAGGTATT
U6 RT and reverse	CGAATTTGCGTGTCATCCT
U6 forward	CTCGCTTCGGCAGCACATA
G6PC forward [13]	CTACTACAGCAACACTTCCGTG
G6PC reverse [13]	GGTCGGCTTTATCTTTCCCTGA
β-actin forward [14]	TGGCACCCAGCACAATGAA
β-actin reverse [14]	CTAAGTCATAGTCCGCCTAGAAGCA

RT: reverse transcription.

**Table 2 ijms-24-16141-t002:** The oligonucleotides sequences for recombinant vectors’ construction.

Name	Sequence (5′-3′)
site 1 wildtype forward	CAATTACTATATTTTAGATGCAATGTGATCTGAAGTTTCTAATTCTGGCCC
site 1 wildtype reverse	TCGAGGGCCAGAATTAGAAACTTCAGATCACATTGCATCTAAAATATAGTAATTGAGCT
site 1 mutant forward	CAATTACTATATTTTAGATGCAATGTGAGAGTAAGTTTCTAATTCTGGCCC
site 1 mutant reverse	TCGAGGGCCAGAATTAGAAACTTACTCTCACATTGCATCTAAAATATAGTAATTGAGCT
site 2 wildtype forward	CGTTACATTTGAAGACATGTCTGCATATCTGAAATTCCAGCCCTAATTAAC
site 2 wildtype reverse	TCGAGTTAATTAGGGCTGGAATTTCAGATATGCAGACATGTCTTCAAATGTAACGAGCT
site 2 mutant forward	CGTTACATTTGAAGACATGTCTGCATAGAGTAAATTCCAGCCCTAATTAAC
site 2 mutant reverse	TCGAGTTAATTAGGGCTGGAATTTACTCTATGCAGACATGTCTTCAAATGTAACGAGCT

## Data Availability

The data generated during the current study are available from the corresponding authors upon reasonable request.

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
