# Peer review of "The Peptide AWRK6 Alleviates Lipid Accumulation in Hepatocytes by Inhibiting miR-5100 Targeting G6PC"

_ijms, 2023, doi:10.3390/ijms242216141_

Round 1
Reviewer 1 Report
Comments and Suggestions for Authors
The authors have clarified part of the molecular mechanism of AWRK6's ability to inhibit fat accumulation, and we believe this is an interesting finding regarding the improvement of NASH/NAFLD.
Please respond to the points you are concerned about below.
1. It's my suggestion that it would be better to place Figure 6 E-G before Figure 6 A.
2. Can you show data on G6PC mRNA and protein expression levels in HepG2 transfected with miR-5100 mimic and additionally supplemented with OA and AWRK6? I felt that presenting these at the end would make the conclusion stronger
Author Response
Dear Editor and Reviewer,
Thank you for your kindly checking and professional comments. The manuscript has been improved following your helpful suggestions. The changes are marked using built-in Track Changes in Microsoft Word. The main modifications and responses to the comments are as follows.
Reviewer 1
The authors have clarified part of the molecular mechanism of AWRK6's ability to inhibit fat accumulation, and we believe this is an interesting finding regarding the improvement of NASH/NAFLD.
Please respond to the points you are concerned about below.
1. It's my suggestion that it would be better to place Figure 6 E-G before Figure 6 A.
Response:
Thank you for your suggestion. Figure 4E-G has been placed before Figure 4A, to make it logical.
2. Can you show data on G6PC mRNA and protein expression levels in HepG2 transfected with miR-5100 mimic and additionally supplemented with OA and AWRK6? I felt that presenting these at the end would make the conclusion stronger
Response:
Thank you for your professional comments. As you pointed out, it would strengthen the conclusion to show G6PC levels upon miR-5100 mimic treatment with OA and AWRK6. Previously, the mRNA and protein levels of G6PC was detected under this condition. However, the changes were not statistically significant. Therefore, we speculate that G6PC is one of the mediators of AWRK6 in alleviating lipid accumulation. Other mechanisms involved are to be further investigated in depth in subsequent studies. It is also possible that we have not found suitable experimental conditions to show the changes in G6PC. We have also discussed this issue you raised in Discussion of the revised manuscript.
Kind regards,
Dianbao Zhang
China Medical University
Reviewer 2 Report
Comments and Suggestions for Authors
In this paper authors reported as the peptide AWRK6 could attenuate lipid accumulation in hepatocytes, in a reducing miR5100 mediated way, becoming a potential peptide for the intervention of NAFLD.
The manuscript is well written, with a correct sequence in the information illustration and the level is appropriate to readership. The topic is very current. Recent studies show that liver derived miRNAs are playing an important role in diagnosis and prognosis in NAFLD and MAFLD.
Minor revision
-Due to the growing importance of the numerous miRNAs in liver dysfunction, authors should better describe, even if in a synthetic way, why the choice fell on mir-5100 and miR-505. In addition, references relating to the selected miRNAs is missing.
-In the fig.1 authors reported how AWRK6 alleviate oleic acid-induced lipid accumulation in HepG2 cells. It would be interesting to report whether significant differences also exist between the groups, not only towards the control (0 mM or OA groups)
-Authors demonstrated the effect of AWRK6 on triglyceride levels. Have they also tested the effect on total cholesterol?
-Finally, the function of miRNAs, exerted through repression of target genes, is tissue specific. Authors should also mention that since it is very difficult to localize the tissue of origin of miRNAs, except for hepatocyte-enriched miR-122 it turns out to be a challenge to explain the disease association to microRNAs regulation.
Author Response
Dear Editor and Reviewer,
Thank you for your kindly checking and professional comments. The manuscript has been improved following your helpful suggestions. The changes are marked using built-in Track Changes in Microsoft Word. The main modifications and responses to the comments are as follows.
Reviewer 2
In this paper authors reported as the peptide AWRK6 could attenuate lipid accumulation in hepatocytes, in a reducing miR5100 mediated way, becoming a potential peptide for the intervention of NAFLD.
The manuscript is well written, with a correct sequence in the information illustration and the level is appropriate to readership. The topic is very current. Recent studies show that liver derived miRNAs are playing an important role in diagnosis and prognosis in NAFLD and MAFLD.
Minor revision
-Due to the growing importance of the numerous miRNAs in liver dysfunction, authors should better describe, even if in a synthetic way, why the choice fell on mir-5100 and miR-505. In addition, references relating to the selected miRNAs is missing.
Response:
Thank you for your helpful comments. In our previous study, we detected miRNAs associated with lipid metabolism using next-generation sequencing and validated them using RT-qPCR, and identified a number of miRNAs that may be associated with AWRK6. the present data are the result of further studies based on these. The reference has been added in the revised version of the manuscript:
Liu, J. MiRNA regulation in the improvement of lipid accumulation in metabolic associated fatty liver disease by AWRK6. Liaoning University 2022, doi: 10.27209/d.cnki.glniu.2022.001436.
-In the fig.1 authors reported how AWRK6 alleviate oleic acid-induced lipid accumulation in HepG2 cells. It would be interesting to report whether significant differences also exist between the groups, not only towards the control (0 mM or OA groups)
Response:
Following your suggestion, the significant differences between the groups has been marked in the revised manuscript. The figure legend has been improved accordingly.
-Authors demonstrated the effect of AWRK6 on triglyceride levels. Have they also tested the effect on total cholesterol?
Response:
Thank you for your helpful comment. In our previous study, the effect of AWRK6 on glycolipid metabolism was investigated at the animal level, and several metrics, including total cholesterol, were examined. In this manuscript, we focused on the triglyceride level. The mechanism of AWRK6 is still in the preliminary stage of exploration, and we will follow your suggestion to study the level of total cholesterol, etc. in subsequent work.
-Finally, the function of miRNAs, exerted through repression of target genes, is tissue specific. Authors should also mention that since it is very difficult to localize the tissue of origin of miRNAs, except for hepatocyte-enriched miR-122 it turns out to be a challenge to explain the disease association to microRNAs regulation.
Response:
Thank you for your professional comments. As you point out, miRNAs have a wide range of roles, can be delivered between different cells and tissues, and are tissue- and cell-specific for functions, making the data on miRNAs difficult to translate to the clinic. This also makes it important to investigate the mode of miRNAs in-depth in order to elucidate the regulatory role of miRNAs in disease development and intervention. This issue you raised has been discussed in Discussion of the revised manuscript.
Kind regards,
Dianbao Zhang
China Medical University
Round 2
Reviewer 1 Report
Comments and Suggestions for Authors
The authors responded appropriately to reviewers' comments.